# Chemical Characterization and Antibacterial Activity of Essential Oil of Medicinal Plants from Eastern Serbia

**DOI:** 10.3390/molecules25225482

**Published:** 2020-11-23

**Authors:** Milica Aćimović, Miroslav Zorić, Valtcho D. Zheljazkov, Lato Pezo, Ivana Čabarkapa, Jovana Stanković Jeremić, Mirjana Cvetković

**Affiliations:** 1Institute of Field and Vegetable Crops Novi Sad, Maksima Gorkog 30, 21000 Novi Sad, Serbia; miroslav.zoric@ifvcns.ns.ac.rs; 2College of Agricultural Sciences, Oregon State University, Corvallis, OR 97331, USA; Valtcho.Jeliazkov@oregonstate.edu; 3Institute of General and Physical Chemistry, University of Belgrade, Studentskitrg 12-16, 11000 Belgrade, Serbia; latopezo@yahoo.co.uk; 4Institute of Food Technology, University of Novi Sad, Bulevar Cara Lazara 1, 21000 Novi Sad, Serbia; ivana.cabarkapa@fins.uns.ac.rs; 5Institute of Chemistry, Technology and Metallurgy, University of Belgrade, Njegoševa 12, 11000 Belgrade, Serbia; jovana_stankovic@hotmail.com (J.S.J.); miracvet13@gmail.com (M.C.)

**Keywords:** *Satureja kitaibelii*, *Thymus serpyllum*, *Origanum vulgare*, *Achillea millefolium*, *Achillea clypeolata*

## Abstract

The objective of this study was to evaluate wild growing *Satureja kitaibelii*, *Thymus serpyllum*, *Origanum vulgare*, *Achillea millefolium* and *Achillea clypeolata* with respect to their essential oil (EO) content, composition and antimicrobial activity. The five species were collected at Mt. Rtanj and the village of Sesalac, Eastern Serbia. The main EO constituents of Lamiaceae plants were *p*-cymene (24.4%), geraniol (63.4%) and germacrene D (21.5%) in *Satureja kitaibelii*, *Thymus serpyllum* and *Origanum vulgare* ssp. *vulgare*, respectively. *A. millefolium* EO had multiple constituents with major ones being camphor (9.8%), caryophyllene oxide (6.5%), terpinen-4-ol (6.3%) and 1,8-cineole (5.6%), while the main EO constituents of *A. clypeolata* were 1,8-cineole (45.1%) and camphor (18.2%). Antimicrobial testing of the EO showed that *Staphylococcus aureus* (Gram-positive) was more sensitive to all of the tested EOs than *Escherichia coli* (Gram-negative). *S. kitaibelii* EO showed the highest antimicrobial activity against both tested bacterial strains. This is the first study to characterize the EO composition and antimicrobial activity of these five medicinal species from Eastern Serbia in comparison with comprehensive literature data. The results can be utilized by the perfumery, cosmetics, food and pharmaceutical industries, but also for healing purposes in self-medication.

## 1. Introduction

The plants of the Lamiaceae and Asteraceae families are widely distributed medicinal plants throughout the world and have been used since ancient times for medicine and food. Historically, medicinal plants from these families have been used for flavoring, food preservation and medicinal purposes, due to their bioactive properties [1,2,3]. Today, pharmaceutical companies are interested in the wide range of beneficial properties of these plants in order to develop modern herbal remedies that would be used either as a replacement or supplement to conventional medicines and for prevention of illnesses [4,5,6]. In addition, medicinal plants are valuable raw material for perfumery and cosmetics, but also for healing purposes in self-medication [7,8,9], as well as in organic agriculture [10,11,12].

In Serbia, the Lamiaceae family includes the biggest number of medicinal plants, followed by Rosaceae and Asteraceae [13]. *Satureja kitaibelii* (Rtanj’s tea), *Thymus serpyllum* (creeping thyme) and *Origanum vulgare* ssp. *vulgare* (common oregano) belong to the Lamiaceae family, while *Achillea millefolium* (yarrow) and *A. clypeolata* (moonshine yarrow) belong to the Asteraceae family. These plants are often utilized in Serbian traditional medicine as herbal teas for the treatment of urinary complaints and digestive disorders (diarrhea, abdominal cramping) [14,15,16]. These plants are also used externally for treating skin and mucous inflammation [17,18].

*Escherichia coli* is the most dominant pathogen causing urinary tract infections [19,20,21] as well as foodborne illnesses [22,23,24], while *Staphylococcus aureus* is the most common cause for foodborne [25,26], skin and soft tissue infectious diseases [27,28,29]. However, over the past decade, the changing pattern of resistance in *E. coli* and *S. aureus* has emphasized the need for new antimicrobial agents [30,31,32]. Because of this, researchers are increasingly turning their attention towards traditional medicine. Therefore, there are a number of reports on some plant extracts and essential oils (EOs) with antimicrobial activity and as a source for antimicrobial agents against food spoilage and pathogens [33,34,35].

The aim of this study was to gather information on five medicinal plants from the Lamiaceae and Asteraceae family from Mt. Rtanj and the village of Sesalac in Eastern Serbia, to determine their EO yield and composition and compare the results with literature data on the same species. A second objective was to assess the antibacterial activity of these EOs against *Staphylococcus aureus* and *Escherichia coli* bacterial strains. Traditionally, the five plant species have been used by Serbian traditional medicine for treatment of various health conditions and diseases as outlined above.

## 2. Results

### 2.1. Chemical Composition

The most abundant compounds in *S. kitaibelii* EO were *p*-cymene (24.4%), limonene (13.5%) and linalool (8.3%), and the most abundant ones in *T. serpyllum* EO were geraniol (63.4%) and nerol (or *cis*-geraniol) (18.9%), while in *O. vulgare* ssp. *vulgare* EO, the most abundant were germacrene D (21.5%), 1,8-cineole (14.2%), sabinene (14.0%) and *trans*-caryophyllene (13.4%). The most abundant constituents of *A. millefolium* EO were camphor (9.8%), caryophyllene oxide (6.5%), terpinen-4-ol (6.3%) and 1,8-cineole (5.6%), while in the *A. clypeolata* EO, they were 1,8-cineole (45.1%) and camphor (18.2%) (Table 1).

### 2.2. Antibacterial Activity

According to the assay, *S. aureus* was more sensitive to the tested EOs than *E. coli*. However, *S. kitaibelii* expressed the most potent activity against both G-positive and G-negative bacteria, followed by *T. serpyllum* and *A. clypeolata*, while *O. vulgare* ssp. *vulgare* showed the weakest antibacterial activity. *A. millefolium* showed weak activity against *E. coli* but was very potent against *S. aureus* (Table 2).

Further investigations will be focused on studying major constituents of *S. kitaibelii*, *T. serpyllum*, *O. vulgare* ssp. *vulgare*, *A. millefolium* and *A. clypeolata* EOs, such as geraniol, 1,8-cineole, *p*-cymene, germacrene D, nerol, camphor, sabinene, limonene, *trans*-caryophyllene, etc., to evaluate concentrations of the components that could be responsible for the antibacterial effect. Additional tests with a larger number of bacteria regarding the synergic potential of EOs will be implemented in our future investigations.

## 3. Discussion

### 3.1. Satureja kitaibelii

*S. kitaibelii* EO constituents from this study and from literature reports [18,36,37,38,39,40] are shown in Table 3. The classification between them according to the content of chemical compounds was performed using HCA analysis (Figure 1a). This analysis showed that differences in EO composition could be separated into several potential clusters (chemotypes): geraniol, *p*-cymene and limonene, with high abundance of linalool and borneol, which is in accordance with a previous study [18].

Taking into account the chemical content range, the correlation network based on components from *S. kitaibelii* EO is shown in Figure 1b. There were strong positive correlations between *p*-cymene and limonene (r = 0.75; *p* < 0.01), as well as between *α*-pinene and sabinene hydrate (r = 0.76; *p* < 0.01). However, the strongest negative correlation was between geraniol and compounds such as *p*-cymene (r = −0.86 *p* < 0.01), limonene (r = −0.74; *p* < 0.01) and borneol (r = −0.66; *p* < 0.01).

The major constituent of extracts and EO used in traditional medicines as an antimicrobial agent is *p*-cymene [41]. However, in this study, *p*-cymene exhibited the weakest antibacterial activity against *S. aureus* and *E. coli* [42]. Contrarily, limonene exhibited moderate antibacterial activity against these bacteria [43], while linalool exhibited strong antimicrobial activity [44]. It is well known that many plants exert their beneficial effects through the additive or synergistic action of several chemical compounds acting at single or multiple target sites [45]. The results from this study suggest that the combination effects of these compounds (*p*-cymene, limonene and linalool) in *S. kitaibelii* EO had antibacterial enhancement (synergistic or additive effects) against tested bacteria (*S. aureus* and *E. coli*). Further investigations into this topic need to be conducted.

### 3.2. Thymus serpyllum

Differentiation between *T. serpyllum* EO samples from this study and from literature reports [46,47,48,49,50,51,52,53,54,55,56,57] (Table 4) was performed using HCA analysis (Figure 2a). Several clusters (potential chemotypes) were established, including thymol, carvacrol, linalool (linalool and linalyl acetate), geraniol (geraniol and geranyl acetate) and terpinene (terpinene and terpinene acetate), as well as a number of multiple-component chemotypes [46,58]. Research conducted on 20 accessions of *T. serpyllum* in Estonia showed the presence of three different chemotypes: *trans*-nerolidol + caryophyllene oxide, *trans*-nerolidol + myrcene and myrcene chemotypes [52]. Furthermore, variability between 16 populations from Poland showed three chemotypes: geranyl acetate + *β*-terpineol + *β*-myrcene, geranyl acetate + *β*-terpineol + borneol and pure linalool chemotype [47].

Data in Figure 2a suggests the presence of 11 potential chemotypes in *T. serpyllum*: geraniol, citronellol, linalool, *α*-terpinyl acetate, *trans*-nerolidol, *trans*-nerolidol + caryophyllene oxide, caryophyllene oxide, carvacrol, thymol, thymol + carvacrol and multiple-component chemotypes. Previous research on *T. vulgaris* described genetically distinct chemotypes that can be distinguished on the basis of the dominant monoterpene produced in the glandular trichomes. It was established that the monoterpene variations in *T. vulgaris* plants may represent an adaptive strategy in relation to the environmental variations, as the different chemotypes showed different geographic and locality distribution [59].

Taking into account the chemical content range, the correlation network based on components from *T. serpyllum* EO is shown in Figure 2b. There were strong negative multiple correlations between sabinene and *β*-caryophyllene, caryophyllene oxide and germacrene D (*p* < 0.01), but all these three compounds are in positive correlations (*p* < 0.01).

Previous research reported that geraniol exhibited good antimicrobial activity and, in combination with antibiotics, would have substantial therapeutic potential against *S. aureus* and *E. coli* infections [60]. Although geraniol and nerol are geometric isomers, they demonstrated equal activities [61]. Taking into account the moderate antibacterial activity against *S. aureus* and *E. coli*, EOs of *T. vulgaris* (with geraniol and nerol) could be used as herbal supplements to conventional therapy.

### 3.3. Origanum vulgare

The classification between *O. vulgare* ssp. *vulgare* EO samples from this study and from literature reports [62,63,64,65,66,67,68,69,70,71,72,73] according to the content of chemical compounds (Table 5) was performed using HCA analysis (Figure 3a), which showed several clusters. *O. vulgare* ssp. *vulgare* collected at Mt. Rtanj can be classified as germacrene D chemotype [65]. The latter authors mentioned three more chemotypes: sabinene, *β*-ocimene and *β*-caryophyllene. The results showed differences among the oregano accessions with respect to morphological traits and chemical constituents of EOs, indicating the existence of infraspecific variations and chemical polymorphism [74]. Chemical composition of aerial parts of *O. vulgare* collected during the flowering season (August 2011), on Mt. Mokra Gora, Southwestern Serbia, showed the presence of sabinene (10.2%), terpinen-4-ol (9.3%), 1,8 cineole (5.8%), *γ*-terpinene (5.6%) and caryophyllene oxide (5.4%) as main compounds [75]. However, another study on *O. vulgare* in Serbia showed the presence of thymol (45%) and carvacrol (37.4%) as the major EO constituents [76].

Correlation network based on *O. vulgare* ssp. *vulgare* EO constituents is shown in Figure 3b. There were two groups of correlations between *O. vulgare* ssp. *vulgare* EO constituents. Compounds in positive correlations were predominantly germacrene D, *β*-caryophyllene, caryophyllene oxide, *α*-terpinene, 1,8-cineole, linalool, sabinene, *p*-cymene, thymol and linalyl acetate. On the other hand, compounds with predominantly negative correlations were *β*-citronellol, carvacrol and *γ*-terpinene (*p* < 0.01), with the last two having a strong positive correlation (r = 0.66, *p* < 0.01).

An antimicrobial effect of synthetic antibiotic in combination with germacrene D showing growth inhibition on *E. coli* and *S. aureus* [77] is noted. Conversely, 1,8-cineole derivatives displayed significant antibacterial activity [78], while *trans*-caryophyllene displayed moderate antibacterial activity [79]. Sabinene exhibited prominent antibiofilm properties against *E. coli* and *S. aureus* providing a novel and effective alternative/complementary approach to counteract chronic infections and the transmission of diseases in clinical settings [80]. The weak antibacterial activity of *O. vulgare* ssp. *vulgare* EO could be attributed to low concentrations of bioactive compounds and their inability to exhibit activity.

### 3.4. Achillea millefolium

*A. millefolium* represents a polyploidic complex of hardly distinguishable species, subspecies, forms and hybrids [81]. However, morphological, chemical and molecular traits as well as PCA analysis showed that the terpenoid variation can be used to explore biogenetic pathways [82]. The differentiation between *A. millefolium* EO from this study and from literature reports [83,84,85,86,87,88,89,90,91,92] (Table 6) was performed using HCA analysis (Figure 4a). Several clusters (potential chemotypes) were established, including camphor, lavandulyl acetate, sabinene, 1,8-cineole, chrysanthenyl acetate, *β*-pinene and chamazulene chemotypes.

According to the major compound of the EO, this *A. millefolium* from Rtanj can be classified as a camphor chemotype. Essential oil of *A. millefolium* from France was also identified as a camphor chemotype (12.8% camphor in EO) [89]. However, in Lithuania, six different chemotypes of *A. millefolium* were found: *β*-pinene, 1,8-cineole, borneol, camphor, nerolidol and chamazulene chemotypes [85]. In Poland, three chemotypes were found based on the determination of the variability between 20 yarrow populations according to content of the most dominant compounds identified in the EO: *β*-pinene, *β*-pinene + chamazulene and 1,8-cineole chemotype [93]. Variations in the essential oil content and composition in commercial samples of yarrow from different European countries were analyzed, and five chemotypes were determined: chamazulene, chamazulene + bornyl acetate, chamazulene + *β*-pinene + *trans*-*β*-caryophyllene, sabinene + 1,8-cineole and *β*-pinene + *α*-terpinyl acetate [92].

The analysis of 28 populations of *A. millefolium* collected from Serbian sites showed that the most dominant compounds were *β*-pinene, sabinene, 1,8-cineole, borneol, *trans*-caryophyllene, lavandulyl acetate and chamazulene [84]. Furthermore, investigations in Serbia showed that the high percentage of oxygenated monoterpenes and absence of azulene in the obtained EO proved that this population was octaploid, whereas the chamazulene chemotype was in the tetraploid population [94]. In addition, collections from saline habitats in Serbia identified three chemotypes: chamazulene + *trans*-caryophyllene + *β*-pinene (15.84 + 8.98 + 8.89%, respectively), lavandulyl acetate + chamazulene + *trans*-caryophyllene (14.88 + 13.89 + 7.57%, respectively) and *trans*-chrysanthenyl acetate + *trans*-caryophyllene + germacrene D (21.33 + 9.53 + 7.07%, respectively) [90].

Taking into account the chemical content range, a correlation network based on components from EO of this species is shown in Figure 4b. The strongest negative multiple correlations were found between *β*-caryophyllene and terpinene-4-ol (r = −0.61; *p* < 0.01) and the strongest positive correlations were between *β*-caryophyllene and chamazulene (r = 0.56; *p* < 0.01).

The *A. millefolium* EO showed antimicrobial activity in vitro against *Streptococcus pneumoniae*, *Clostridium perfringens*, *C. albicans*, *C. krusei*, *Mycobacterium smegmatis* and *Acinetobacter lwoffii* [88]. Additionally, in vitro antibacterial activity against nine Gram positive and negative bacteria (*S. epidermidis*, *S. aureus*, *B. cereus*, *E. faecalis*, *E. coli*, *P. aeruginosa*, *K. pneumoniae*, *S. typhimurium* and *Shigella dysenteria*) demonstrated that *A. millefolium* EO can potentially be used for controlling certain bacteria that cause many infectious diseases, but its effectiveness varied in different regions because of the differences in EO composition [87].

### 3.5. Achillea clypeolata

The main compounds in the *A. clypeolata* EO from Mt. Rtanj (collected in July 1996) were 1,8-cineole (38.6%) and camphor (19.9%) [95]. The concentration of 1,8 cineole in *A. clypeolata* in this study was a bit higher. This variation in the chemical composition could be attributed to the weather conditions during the year, collection time, population and exposition. Because this is an endemic species in the Balkan region, there were very few studies on its EO composition. *A. clypeolata* of Serbian origin contained 1,8-cineole as the dominant compound [95,96], and 1,8-cineole as a potential chemotype of this species was confirmed by this study. However, *A. clypeolata* grown in a botanical garden in Italy could be characterized as *β*-pinene chemotype [97].

The differentiation between *A. clypeolata* EO samples (Table 7) was performed using HCA analysis (Figure 5a). Only two clusters (potential chemotypes) were established, including 1,8-cineole and *β*-pinene chemotype.

Taking into account the chemical content range, a correlation network based on components from EO of this species is shown in Figure 5b. There were strong negative correlations between terpinen-4-ol and *β*-pinene as well as *β*-pinene and borneol. Positive correlations are noted between terpinen-4-ol and borneol, as well as multiple correlation between *allo*-aromadendrene, *epi*-*α*-cadinol and *α*-cadinol.

*A. clypeolata* has not been investigated thoroughly yet, with the exception of its antioxidant and antimicrobial properties [96,98]. In this study, the EO of *A. clypeolata* exhibited the strongest activity against *K. pneumoniae* and *P. aeruginosa*, even stronger than the antibiotic Ampicilin, which was used as the standard for comparison. *A. clypeolata* EO showed lower activity against Gram-positive *S. aureus*, but still stronger than Ampicilin, while *E. coli* was the most resistant to the oil [96]. Indeed, previous study on *Cinnamomum longepaniculatum* leaf EO has shown that its compounds have excellent antibacterial activities, and the antibacterial mechanism of 1,8-cineole against *E. coli* and *S. aureus* might be attributed to its hydrophobicity [99].

## 4. Material and Methods

### 4.1. Plant Material

The plant species were collected at full flowering stage in July 2018. *Satureja kitaibelii* Wierzb. ex Heuff., *Origanum vulgare* L. ssp. *vulgare*, *Achillea millefolium* L. *sensu lato* and *A. clypeolata* Sibth. & Sm were collected on Mt. Rtanj and *Thymus serpyllum* L. was collected in the village of Sesalac. The aboveground parts were harvested by cutting them manually at around 2–3 cm above the soil surface and were then dried in a shady well-aerated place to a constant mass.

Voucher specimens were identified by Dr. Milica Rat and deposited at the Herbarium BUNS, the University of Novi Sad, Faculty of Sciences, Department of Biology and Ecology as *S. kitaibelii* Wierzb. ex Heuff. (2-1442), *T. serpyllum* L. (2-1444), *O. vulgare* L. ssp. *vulgare* (2-1450), *A. millefolium* L. *sensu lato* (2-1449) and *A. clypeolata* Sibth. & Sm (2-1448).

### 4.2. Essential Oil (EO) Extraction

Clevenger apparatus was used to extract the essential oil from the air-dried aboveground parts of each sample in three replications. According to the European Pharmacopoeia, 30 g each of the drug plants from the Lamiaceae family (*S. kitaibelii*, *T. serpyllum* and *O. vulgare* ssp. *vulgare*) and 400 mL of water and 20 g each of cut drug from the Asteraceae family (*A. millefolium* and *A. clypeolata*) and 500 mL of water were placed separately in 1000 mL round bottom flasks and distilled for 3 h [100].

The EO yield as an average of three replications was *S. kitaibelii* (0.09%), *T. serpyllum* (0.26%), *O. vulgare* ssp. *vulgare* (0.12%), *A. millefolium* (0.16%) and *A. clypeolata* (0.04%). The obtained EOs were dried over anhydrous NaSO_4_ and stored in the dark at 4 °C for further analysis.

### 4.3. Essential Oil (EO) Analysis

Gas chromatography–mass spectrometry (GC-MS) was performed using an HP 5890 coupled with an HP 5973 MSD fitted with an HP-5MS capillary column (conditions in detail described by Aćimović et al. [101]). The components were identified based on their linear retention index relative to C8–C32 n-alkanes, in comparison with data reported in the literature (Wiley and NIST databases). The percentage (relative) of the identified compounds was computed from the GC peak area.

### 4.4. Association among Chemical Compounds

The association of the chemical compounds of EOs was estimated using the Spearman non-parametric correlation coefficient. In order to further analyze and represent the associations among the chemical compounds, a correlation network graph [102] was used. In this type of graph, the chemical compound variables are represented by the nodes, which are connected by the edge whose width is directly proportional to the strength of the correlation coefficient. The correlation network graph is easier to interpret than a numerical correlation matrix, and the pattern of the correlations, i.e., the clusters of the correlated variables, can be visually identified [103].

A data set, composed of 16 *S. kitaibelii*, 54 *T. serpyllum*, 53 *O. vulgare* ssp. *vulgare*, 63 *A. millefolium* and 4 *A. clypeolata* samples and 13 variables (the main EO constituents), were depicted using a correlation network graph for visualization correlations between chemical compounds from EOs (Spearman’s rank order correlation coefficient, r). Non-significant correlations were removed from the correlation network graph.

### 4.5. Antibacterial Activity

The antimicrobial activities of the tested EOs were investigated using American Type Culture Collection test strains of *Escherichia coli* (ATCC 8739) and *Staphylococcus aureus* (ATCC 25923). Strains were cultured on Tryptone Soya Agar (TSA) and incubated at 37 °C for 24 h. Isolated colonies were picked and transferred to 5 mL of Tryptone Soy Broth (TSB) and incubated at 37 °C for 18 h. The density of the suspensions used for tests was adjusted to 0.5 Mc Farland units (~1–2 × 10^8^ CFU/mL) using a densitometer DEN-1 and standard plate counts. Efficacy of EOs on microorganisms was determined according to the CLSI (2018) [104] with slight modifications [105].

Tested EO was dissolved with sterile water supplemented with 0.05% Tween 80 and added to 96-well microtiter plates at concentrations from 200 to 0.781 μL/mL (the final concentration in microtiter plates was from 20 to 0.078 μL/mL). The 160 µL in Mueller-Hinton Broth (MHB) were also added to each well, and in the end, 20 µL of overnight bacterial cultures suspensions were inoculated. Plates were incubated at 37 °C for 24 h. After incubation, 20 µL of the resazurin solution (0.01%) were added to each well, and the plates were further incubated at 37 °C for 24 h in darkness. A change of color from blue (oxidized-resazurin remained unchanged) to pink (reduced) indicated the growth of bacteria.

Referring to the results of the minimal inhibitory concentration (MIC), the wells showing a complete absence of growth were identified and 100 µL of the solutions from each well were transferred to plate count agar plates (PCA) and incubated at 37 °C for 24 h. The minimal bactericidal concentration (MBC) was defined as the lowest concentration of the EOs at which 99.9% of the inoculated microorganisms were killed.

### 4.6. Statistical Analysis

Experimental results were expressed as mean ± standard deviation, with 10 repetitions for microbiological analyses. Analysis of variance (ANOVA) for comparison of sample means and a post-hoc Tukey’s HSD test was used to analyze variations in observed parameters among the samples.

Hierarchical cluster analysis (HCA) was used to evaluate intra- and interpopulation variability and differentiation of EO constituents of *S. kitaibelii*, *T. serpyllum*, *O. vulgare* ssp. *vulgare*, *A. millefolium* and *A. clypeolata* in samples collected in different locations and/or taken from literature reports. Data was analyzed using StatSoft Statistica 12.

## 5. Conclusions

This is the first study to characterize EO composition and antimicrobial activity of the five medicinal species from Eastern Serbia against comprehensive literature data. This study demonstrated that *S. kitaibelii* plants from Mt. Rtanj belonged to the *p*-cymene chemotype, *T. serpyllum* plants belonged to the geraniol chemotype, while the *O. vulgare* ssp. *vulgare* plants belonged to the germacrene D chemotype. The *A. millefolium* were recorded as a multicomponent chemotype, while *A. clypeolata* belonged to 1,8-cineole (45.1%) chemotype. Furthermore, the *S. kitaibelii* EO was demonstrated to be a promising agent against *S. aureus* and *E. coli* bacterial strains. The chemical composition of studied EOs particularly focuses on the main EO constituents, which are assumed to be responsible for the observed antibacterial activity. Further investigations will be focused on studying major constituents of these EOs to evaluate concentrations of the components that could be responsible for the observed effect. Additional tests with larger numbers of bacteria regarding the synergic potential of EOs will be implemented in our future investigations.

## Figures and Tables

**Figure 1 molecules-25-05482-f001:**
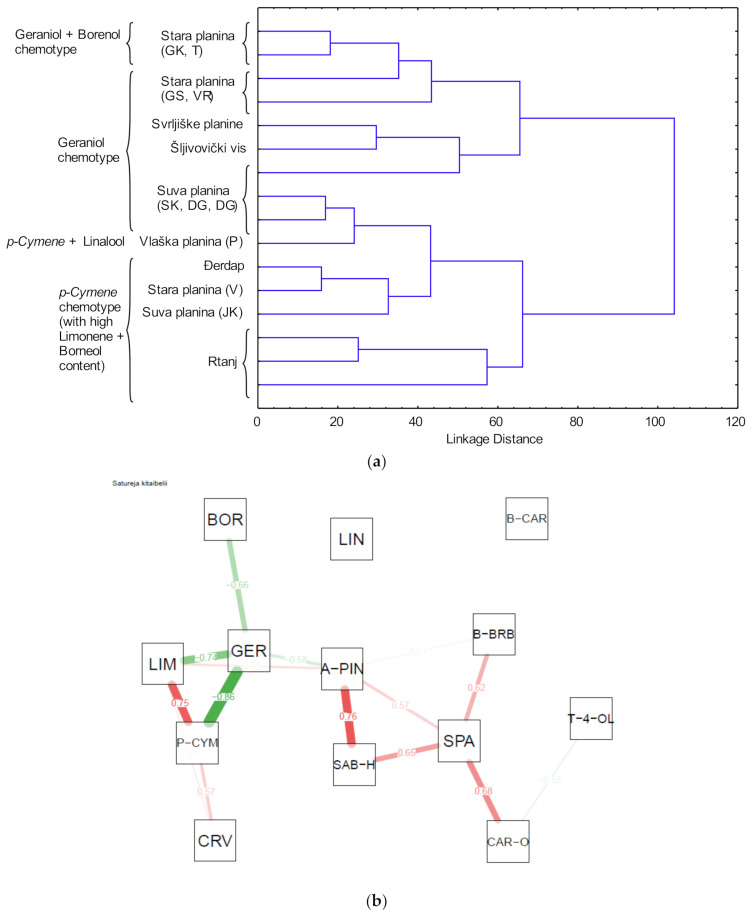
(**a**) Dendogram of the EO constituents of *S. kitaibelii* from this study and from literature reports (the samples are marked according to Table 3); (**b**) correlation network based on *S. kitaibelii* EO constituents (P-CYM—*p*-cymene; CRV—Carvacrol; A-PIN—*α*-pinene; LIM—limonene; SAB-H—*cis*-sabinene hydrate; T-4-OL—terpinen-4-ol; LIN—linalool; BOR—borneol; GER—geraniol; B-BRB—*β*-bourbonene; B-CAR—*β*-caryophyllene; SPA—spathulenol; CAR-O—caryophyllene oxide).

**Figure 2 molecules-25-05482-f002:**
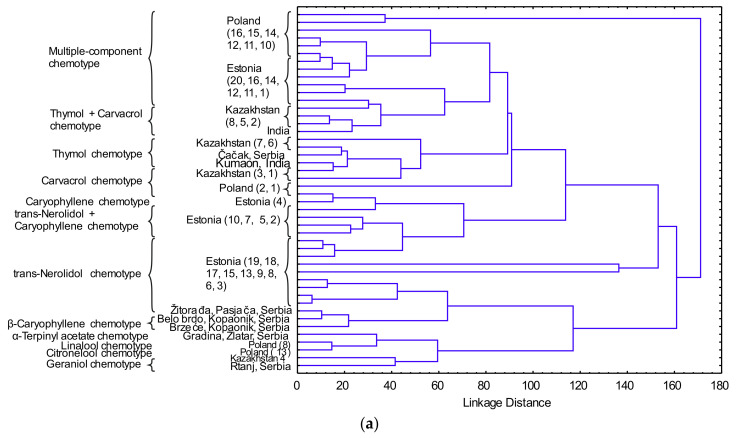
(**a**) Dendogram of the EO constituents of *T. serpyllum* from this study and from literature reports (the samples are marked according to Table 4); (**b**) correlation network based on *T. serpyllum* EO constituents (THY—thymol; CRV—carvacrol; MYR—myrcene; G-TER—*γ*-terpinene; NER—nerol; GER—geraniol; LIN—linalool; CIT—citronellol; A-TRP—*α*-terpinyl acetate; B-CAR—*β*-caryophyllene; GRM—germacrene; CAR-O—caryophyllene oxide; T-NER—*trans*-nerolidol).

**Figure 3 molecules-25-05482-f003:**
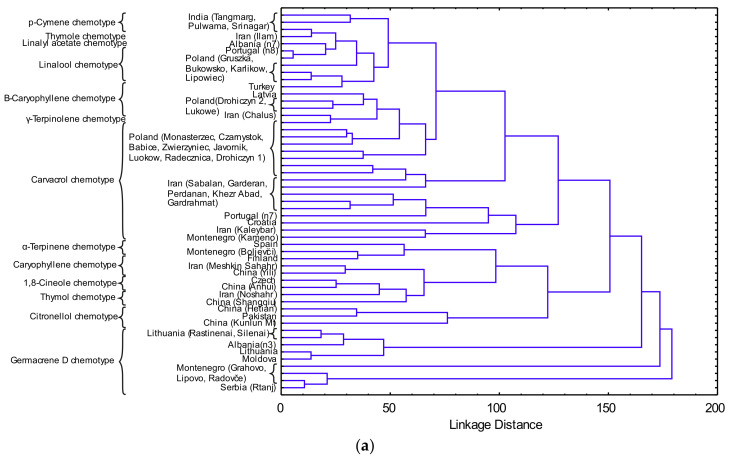
(**a**) Dendogram of the EO constituents of *O. vulgare* from this study and from literature reports (the samples are marked according to Table 5); (**b**) correlation network based on *O. vulgare* EO constituents (P-CYM—*p*-cymene; THY—thymol; CRV—carvacrol; SAB—sabinene; G-TER—*γ*-terpinene; 1,8-CIN—1,8-cineole; LIN—linalool; A-TER—*α*-terpineol; B-CIT—*β*-citronellol; LIN-A—linalyl acetate; B-CAR—*β*-caryophyllene; GER-D—germacrene D; CAR-O—caryophyllene oxide).

**Figure 4 molecules-25-05482-f004:**
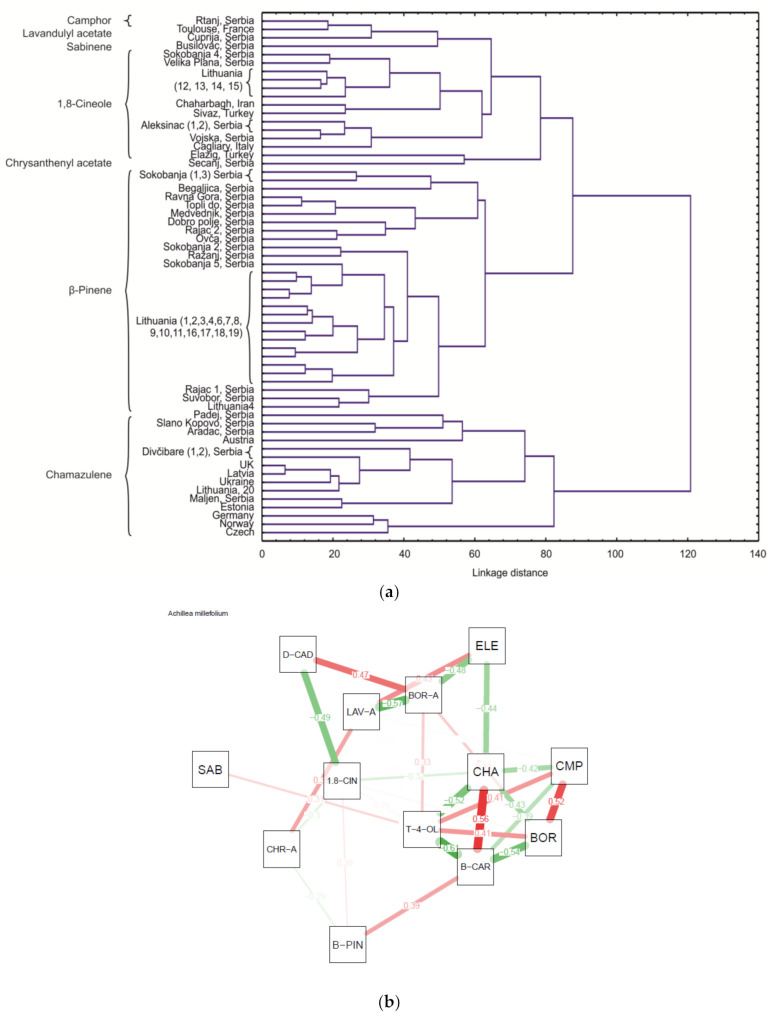
(**a**) Dendogram of the *A. millefolium* EO constituents from this study and from literature reports (the samples are marked according to Table 6); (**b**) correlation network based on *A. millefolium* EO constituents (SAB—sabinene; B-PIN—*β*-pinene; CMP—camphor; 1,8-CIN—1,8-cineole; BOR—borneole; T-4-OL—terpinene-4-ol; BOR-A—bornyl acetate; CHR-A—chrysanthenyl acetate; LAV-A—lavandulyl acetate; CHA—chamazulene; B-CAR—*β*-caryophyllene; D-CAD—δ-cadinene; ELE—elemol.

**Figure 5 molecules-25-05482-f005:**
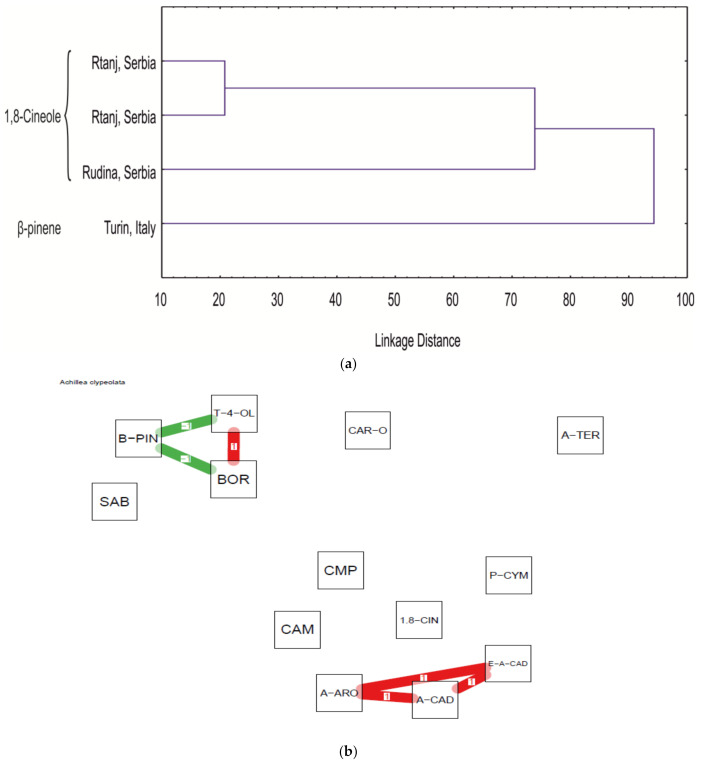
(**a**) Dendogram of the EO constituents of *A. clypeolata* from this study and from literature reports (the samples are marked according to Table 7); (**b**) correlation network based on *A. clypeolata* EO constituents (CAM—camphene; SAB—sabinene; B-PIN—*β*-pinene; P-CYM—p-cymene; 1,8-CIN—1,8-cineole; CMP—camphor; BOR—borneol; T-4-OL—terpinene-4-ol; A-TER—*α*-terpineol; A-ARO—allo-aromadendrene; CAR-O—caryophyllene oxide; E-A-CAD—epi-*α*-cadinol; A-CAD—*α*-cadinol.

**Table 1 molecules-25-05482-t001:** Chemical composition of investigated essential oils (EOs).

	*S. kitaibelii*		*T. serpyllum*		*O. vulgare* ssp. *vulgare*		*A. millefolium*		*A. clypeolata*	
Compound	RI	%	RI	%	RI	%	RI	%	RI	%
Tricyclene	-	nd	-	nd	-	nd	-	nd	927	0.1
1-Octen-3-ol	-	nd	981	0.2	-	nd	-	nd	-	nd
*α*-Thujene	919	0.2	-	nd	915	0.1	928	0.2	929	0.1
*α*-Pinene	927	2.5	-	nd	926	1.5	935	0.7	936	1.1
Camphene	945	1.1	-	nd	944	0.9	949	0.4	950	2.3
Thuja-2,4(10)-diene	-	nd	-	nd	-	nd	-	nd	955	0.1
Sabinene	966	0.2	-	nd	967	14.0	974	2.8	975	0.3
*β*-Pinene	971	0.5	-	nd	971	3.9	978	1.5	979	1.4
Myrcene	988	0.8	-	nd	-	nd	-	nd	-	nd
dehydro-1,8-Cineole	-	nd	-	nd	-	nd	991	0.2	993	0.1
3-Octanol	994	0.1	-	nd	-	nd	-	nd	-	nd
*α*-Phellandrene	1005	0.2	-	nd	-	nd	-	nd	-	nd
*α*-Terpinene	1015	0.9	-	nd	-	nd	1015	0.4	1016	0.6
*p*-Cymene	1022	24.4	-	nd	-	nd	1022	0.9	1022	3.1
*β*-Phellandrene	-	nd	-	nd	1026	1.6	-	nd	-	nd
Limonene	1026	13.5	-	nd	-	nd	1027	0.1	-	nd
1,8-Cineole	1028	1.0	-	nd	1028	14.2	1028	5.6	1028	45.1
*cis*-*β*-Ocimene	1032	2.3	-	nd	1033	6.8	-	nd	-	nd
*trans*-*β*-Ocimene	1042	1.8	-	nd	1043	4.5	-	nd	-	nd
*γ*-Terpinene	1052	3.3	-	nd	1052	0.4	1052	1.1	1053	1.2
*cis*-Sabinene hydrate	1060	2.4	-	nd	-	nd	1060	0.4	-	nd
Terpinolene	1081	0.4	-	nd	-	nd	1080	0.3	-	nd
*p*-Mentha-2,4(8)-diene	-	nd	-	nd	-	nd	-	nd	1081	0.2
Linalool	1092	8.3	1098	0.2	-	nd	1092	4.2	1092	0.4
*n*-Nonanal	-	nd	-	nd	-	nd	1096	0.2	-	nd
*cis*-*p*-Menth-2-en-1-ol	1112	0.3	-	nd	-	nd	1113	0.2	1114	0.2
Chrysanthenone	-	nd	-	nd	-	nd	1116	0.2	-	nd
Nerol oxide	-	nd	1155	0.1	-	nd	-	nd	-	nd
*α*-Campholenal	1118	0.1	-	nd	-	nd	-	nd	1119	0.2
*trans*-Pinocarveol	-	nd	-	nd	-	nd	1132	0.1	-	nd
*trans*-*p*-Menth-2-en-1ol	-	nd	-	nd	-	nd	1133	0.1	-	nd
Camphor	1137	0.4	-	nd	-	nd	1138	9.8	1136	18.2
Pinocarvone	-	nd	-	nd	-	nd	1155	0.7	-	nd
*cis*-Chrysanthenol	-	nd	-	nd	-	nd	-	nd	1156	0.3
Borneol	1160	4.9	1167	0.1	1160	1.2	1159	1.6	1159	2.7
*δ*-Terpineol	-	nd	-	nd	-	nd	-	nd	1161	0.6
*cis*-Pinocamphone	-	nd	-	nd	-	nd	1167	0.1	-	nd
Menthol	1167	0.1	-	nd	-	nd	-	nd	-	nd
Terpinen-4-ol	1172	3.4	-	nd	1172	0.3	1173	6.3	1172	2.8
Thuj-3-en-10-al	-	nd	-	nd	-	nd	1179	0.1	-	nd
*p*-Cymen-8-ol	1181	0.2	-	nd	-	nd	-	nd	-	nd
*trans*-*p*-Mentha-1(7),8-dien-2-ol	1184	0.1	-	nd	-	nd	-	nd	-	nd
*α*-Terpineol	1187	0.4	-	nd	1187	0.7	1186	1.3	1186	2.4
Myrtenol	-	nd	-	nd	-	nd	1189	0.2	-	nd
Myrtenal	-	nd	-	nd	-	nd	1191	0.4	1192	0.3
*cis*-Dihydro carvone	1193	0.7	-	nd	-	nd	-	nd	-	nd
*trans*-Dihydro carvone	1201	1.8	-	nd	-	nd	-	nd	-	nd
*trans*-Carveol	1216	0.3	-	nd	-	nd	-	nd	-	nd
Isobornylformate	1225	0.1	-	nd	-	nd	-	nd	-	nd
*cis*-Carveol	-	nd	-	nd	-	nd	1226	0.1	-	nd
*trans*-Chrysanthenyl acetate	-	nd	-	nd	-	nd	1231	0.2	-	nd
Nerol	-	nd	1233	18.9	-	nd	-	nd	-	nd
Cumin aldehyde	1237	0.2	-	nd	-	nd	1235	0.1	-	nd
Thymol, methyl ether	-	nd	1239	0.2	-	nd	-	nd	-	nd
Carvacrol, methyl ether	1241	0.7	-	nd	-	nd	-	nd	-	nd
Neral	-	nd	1245	0.7	-	nd	-	nd	-	nd
*cis*-Chrysanthenyl acetate	-	nd	-	nd	-	nd	1257	0.4	-	nd
Geraniol	-	nd	1258	63.4	-	nd	-	nd	-	nd
Geranial	-	nd	1275	1.2	-	nd	-	nd	-	nd
Bornyl acetate	-	nd	-	nd	1286	0.1	1282	0.4	-	nd
Thymol	1291	0.2	1298	0.1	-	nd	1288	0.6	1290	0.9
Carvacrol	1299	2.3	-	nd	-	nd	1298	1.1	1300	0.5
*p*-Mentha-1,4,-dien-7-ol	-	nd	-	nd	-	nd	1325	0.2	-	nd
Methyl geranate	-	nd	1328	0.2	-	nd	-	nd	-	nd
*trans*-Carvyl acetate	-	nd	-	nd	-	nd	1334	0.1	-	nd
Eugenol	-	nd	-	nd	-	nd	1354	0.1	-	nd
*cis*-Carvyl acetate	-	nd	-	nd	-	nd	1359	0.1	-	nd
Neryl acetate	-	nd	1368	0.3	-	nd	-	nd	-	nd
*α*-Copaene	1371	0.3	-	nd	-	nd	1373	0.1	-	nd
*β*-Bourbonene	1383	0.9	-	nd	1384	1.9	1387	0.1	-	nd
Geranyl acetate	-	nd	1388	4.7	-	nd	-	nd	-	nd
*β*-Elemene	1390	0.2	-	nd	-	nd	-	nd	-	nd
*cis*-Jasmone	-	nd	-	nd	-	nd	1395	0.1	-	nd
Methyl eugenol	-	nd	-	nd	-	nd	1401	0.1	-	nd
*trans*-Caryophyllene	1418	2.8	1425	4.6	1420	13.4	1418	4.7	1417	0.5
*β*-Copaene	1428	0.2	-	nd	-	nd	-	nd	-	nd
*cis*-*β*-Farnesene	-	nd	-	nd	-	nd	1441	0.1	-	nd
*α*-Humulene	1453	0.1	1459	0.2	1454	2.1	1441	0.1	-	nd
*trans*-*β*-Farnesene	-	nd	-	nd	-	nd	1455	0.1	-	nd
*allo*-Aromadendrene	-	nd	-	nd	-	nd	-	nd	1458	1.3
9-*epi*-*trans*-Caryophyllene	-	nd	-	nd	1461	0.5	1459	0.4	-	nd
*γ*-Muurolene	1476	0.1	-	nd	-	nd	1473	0.3	-	nd
Germacrene D	1481	3.6	1487	0.3	1486	21.5	1480	2.9	1484	1.2
*β*-Selinene	-	nd	-	nd	-	nd	1484	0.2	-	nd
*trans*-Muurola-4(14),5-diene	-	nd	-	nd	-	nd	1490	0.1	-	nd
*epi*-Cubebol	-	nd	-	nd	-	nd	1493	0.5	-	nd
Bicyclogermacrene	1497	1.4	-	nd	1497	1.7	-	nd	-	nd
*β*-Bisabolene	1508	1.0	1515	2.0	-	nd	-	nd	-	nd
(*trans*, *trans*)-*α*-Farnesene	-	nd	-	nd	1509	0.7	-	nd	-	nd
*γ*-Cadinene	-	nd	-	nd	-	nd	1512	1.1	1512	1.2
*δ*-Cadinene	1523	0.2	-	nd	1524	1.1	-	nd	1522	0.2
*α*-Calacorene	-	nd	-	nd	-	nd	1540	0.3	-	nd
Elemol	-	nd	-	nd	-	nd	1546	0.7	-	nd
*trans*-Nerolidol	-	nd	-	nd	-	nd	1560	0.1	-	nd
Geranyl butanoate	-	nd	1565	0.2	-	nd	-	nd	-	nd
Germacrene D-4-ol	-	nd	-	nd	1575	0.8	-	nd	-	nd
*ar*-Tumerol	-	nd	-	nd	-	nd	1575	0.6	-	nd
Spathulenol	1575	2.0	-	nd	1577	0.6	-	nd	1573	0.4
Caryophyllene oxide	1581	2.9	1587	0.9	1583	3.9	1580	6.5	1582	3.2
Viridiflorol	-	nd	-	nd	-	nd	1587	0.5	-	nd
Salvial-4(14)-en-1-one	1591	0.1	-	nd	-	nd	-	nd	-	nd
Ledol	-	nd	-	nd	-	nd	1596	0.1	-	nd
Geranyl isovalerate	-	nd	1606	0.2	-	nd	-	nd	-	nd
Humulene epoxide II + *β*-Oplopenone	-	nd	-	nd	1606	0.3	1603	0.6	1603	0.2
Muurola-4,10(14)-dien-1-*β*-ol	-	nd	-	nd	-	nd	-	nd	1622	0.3
*γ*-Eudesmol	-	nd	-	nd	-	nd	1626	1.5	-	nd
Caryophylla-4(12),8(13)-dien-5-*α*-ol	-	nd	1640	0.4	-	nd	1631	1.8	1630	0.7
*epi*-*α*-Cadinol	-	nd	-	nd	-	nd	-	nd	1635	2.4
*epi*-*α*-Murrolol (=tau-muurolol)	-	nd	-	nd	1640	0.4	1636	3.3	-	nd
*β*-Eudesmol	-	nd	-	nd	-	nd	1646	2.4	-	nd
*α*-Cadinol	-	nd	-	nd	1654	0.9	1648	1.2	1648	1.5
14-hydroxy-9-*epi*-*trans*-Caryophyllene	1669	0.3	-	nd	-	nd	-	nd	-	nd
*α*-Bisabolol	-	nd	-	nd	-	nd	1678	0.3	-	nd
Germacra-4(15),5,10(14) -trien-1-*α*-ol	-	nd	-	nd	-	nd	1681	1.7	-	nd
Eudesma-4(15),7-dien-1-*β*-ol	1684	0.4	-	nd	-	nd	-	nd	-	nd
Curcuphenol	-	nd	-	nd	-	nd	1711	0.2	-	nd
2-*cis*,6-*trans* Farnesol	-	nd	-	nd	-	nd	1715	0.2	-	nd
Chamazulene	-	nd	-	nd	-	nd	1724	0.1	-	nd
6R,7R-Bisabolone	-	nd	-	nd	-	nd	1739	1.2	-	nd
*β*-Costol	-	nd	-	nd	-	nd	1761	0.3	-	nd
6,10,14-trimethyl-2-Pentadecanone	-	nd	-	nd	-	nd	1840	0.2	-	nd
Heptadecanal	-	nd	-	nd	-	nd	1913	0.1	-	nd
Heneicosane	-	nd	-	nd	-	nd	2100	0.2	-	nd
Phytol	-	nd	-	nd	-	nd	2123	0.1	-	nd
(*cis*, *cis*)-9,12-Octadecadienoic acid	-	nd	-	nd	-	nd	2148	0.1	-	nd
*trans*-Geranylgeraniol	-	nd	-	nd	-	nd	2172	0.3	-	nd
Tricosane	-	nd	-	nd	-	nd	2301	0.6	-	nd
Pentacosane	-	nd	-	nd	-	nd	2497	0.4	-	nd
Heptacosane	2701	0.1	-	nd	-	nd	2701	0.2	-	nd
Nonacosane	2896	0.1	2895	0.1	-	nd	2909	0.3	-	nd
**Total**		**96.8**		**99.2**		**100.0**		**80.5**		**98.3**

RI—retention index on a HP-5MS column; nd—not detected.

**Table 2 molecules-25-05482-t002:** Antimicrobial activity of tested EO against *Escherichia coli* and *Staphylococcus aureus* (µL/mL).

	*Escherichia coli* (ATCC8739) G-Negative		*Staphylococcus aureus* (ATCC25923) G-Positive	
	MIC	MBC	MIC	MBC
*S. kitaibelii*	0.156 ± 0.013 ^a^	0.312 ± 0.023 ^a^	0.078 ± 0.002 ^a^	0.156 ± 0.005 ^a^
*T. serpyllum*	20.00 ± 0.52 ^c^	10.00 ± 0.05 ^b^	2.50 ± 0.19 ^c^	5.00 ± 0.21 ^c^
*O. vulgare* ssp. *vulgare*	20.00 ± 0.43 ^c^	>20.00 ^d^	10.00 ± 0.03 ^d^	20.00 ± 0.43 ^d^
*A. millefolium*	20.00 ± 0.23 ^c^	>20.00 ^d^	1.25 ± 0.10 ^b^	2.50 ± 0.17 ^b^
*A. clypeolata*	10.00 ± 0.24 ^b^	20.00 ± 0.41 ^c^	2.50 ± 0.09 ^c^	5.00 ± 0.04 ^c^

Means in the same column with different superscript are statistically different, according to Tukey’s HSD test (*p* ≤ 0.05); *n* = 10 repetitions; MIC—minimal inhibitory concentration; MBC—minimal bactericidal concentration.

**Table 3 molecules-25-05482-t003:** The essential oil constituents of *S. kitaibelii* from this study and from literature reports.

Locality	Ref.	p-Cymene	Carvacrol	*α*-Pinene	Limonene	cis-Sabinene hydrate	Terpinen-4-ol	Linalool	Borneol	Geraniol	*β*-Bourbonene	*β*-Caryophyllene	Spathulenol	Caryophyllene oxide
Rtanj	TS	24.4	2.3	2.5	13.5	2.4	3.4	8.3	4.9	0.0	0.9	0.0	2.0	2.9
Rtanj	[36]	33.6	14.1	1.1	8.5	5.5	1.2	3.0	3.6	0.0	0.0	0.0	2.1	2.3
Rtanj	[37]	22.3	2.9	4.3	11.4	1.1	2.3	5.9	7.6	0.0	0.0	1.5	2.4	1.7
Suva planina (JK)	[38]	20.9	0.0	6.0	16.0	8.2	3.8	1.6	9.8	0.0	0.0	0.0	0.0	0.0
Staraplanina (V)	[39]	34.2	0.0	0.0	10.5	0.0	0.0	5.0	10.4	0.0	0.0	0.0	0.0	3.0
Đerdap	[18]	21.9	0.0	2.4	8.0	9.0	3.8	0.0	7.7	0.0	1.5	2.2	2.9	5.9
Vlaškaplanina (P)	[18]	16.9	0.0	3.2	7.4	3.2	2.5	22.2	7.6	0.0	1.9	1.2	1.2	1.6
Suva planina (DG)	[40]	4.3	0.0	0.6	7.9	5.1	0.0	5.8	3.7	28.1	0.0	0.0	2.1	5.0
Suva planina (DG)	[39]	7.7	0.0	0.0	6.1	0.0	0.0	12.0	0.0	23.2	0.0	0.0	0.0	4.3
Suva planina (SK)	[40]	1.4	0.0	1.1	4.3	3.8	0.0	14.8	1.8	30.3	0.3	0.0	2.8	5.2
Šljivovički vis	[39]	4.4	0.0	0.0	5.7	0.0	0.0	9.5	5.4	23.2	0.0	0.0	0.0	0.0
Svrljiškeplanine	[39]	9.3	0.0	0.0	7.9	0.0	10.8	10.7	0.0	13.4	0.0	0.0	0.0	0.0
Staraplanina (VR)	[40]	1.4	0.0	0.8	5.9	0.7	0.0	5.0	2.4	24.0	0.0	0.0	1.5	4.4
Staraplanina (GS)	[39]	7.5	0.0	0.0	0.0	0.0	4.1	6.2	0.0	29.7	0.0	4.5	0.0	0.0
Staraplanina (T)	[18]	5.5	0.0	2.0	5.2	2.0	3.2	0.3	7.7	12.0	5.9	4.2	5.8	3.8
Staraplanina (GK)	[39]	6.6	0.0	0.0	0.0	0.0	10.3	6.5	9.4	13.0	0.0	0.0	0.0	0.0

Ref.—reference; TS—this study; JK—Jelasnicka Klisura; V—Vetren; P—Poganovo; DG—Devojacki Grob; SK—Sicevacka Klisura; VR—Visoka Rzana; GS—Gornja Sokolovica; T—Temsica; GK—Golemei Kamen.

**Table 4 molecules-25-05482-t004:** The essential oil constituents of other *T. serpyllum* reported in the literature.

Locality	Ref.	Myrcene	*γ*-Terpinene	Nerol	Geraniol	Thymol	Carvacrol	*α*-Terpinyl Acetate	*β*-Caryophyllene	Linalool	Citronellol.	Trans-Nerolidol	Germacrene D-4-ol	Caryophyllene Oxide
Serbia, Rtanj	TS	0.0	0.0	18.9	63.4	0.1	0.0	0.0	4.6	0.2	0.0	0.0	0.3	0.9
Kazakhstan 4	[46]	0.0	1.4	2.8	55.9	5.1	1.2	0.0	0.0	0.0	0.0	0.0	0.0	0.0
Poland 13	[47]	1.0	0.6	3.9	0.0	0.2	1.0	0.0	0.0	0.7	23.9	0.3	0.0	1.0
Poland 8	[47]	0.6	0.5	0.0	1.0	0.1	0.2	0.0	0.0	56.6	0.7	0.3	0.0	0.2
Serbia, Zlatar (G)	[48]	8.2	0.2	0.0	0.0	0.2	0.0	66.2	2.7	0.2	0.0	0.0	0.5	0.6
Serbia, Kopaonik (B)	[49]	3.8	1.8	0.0	0.0	0.1	0.0	0.0	33.3	0.0	0.0	0.1	0.1	1.7
Serbia, Kopaonik (BB)	[50]	0.0	1.4	0.0	0.0	5.6	0.0	0.0	27.7	1.2	0.0	2.4	1.1	1.3
Serbia, Pasjača (Ž)	[51]	1.6	1.5	0.0	1.4	7.3	0.6	1.0	2.8	0.7	0.0	24.2	16.0	1.1
Estonia 3	[52]	0.8	0.0	0.0	0.0	0.5	0.3	0.0	3.8	0.0	0.0	52.0	3.3	4.2
Estonia 6	[52]	0.2	0.0	0.0	0.0	2.9	1.7	0.0	3.1	0.0	0.0	49.5	3.8	11.2
Estonia 8	[52]	0.0	0.0	0.0	0.0	1.0	0.3	0.0	1.8	0.0	0.0	70.1	2.8	4.5
Estonia 9	[52]	0.3	0.0	0.0	0.0	0.5	0.3	0.0	5.7	0.0	0.0	20.5	5.3	6.4
Estonia 13	[52]	8.6	0.0	0.0	0.0	1.4	2.0	0.0	4.6	0.0	0.0	24.3	7.9	2.5
Estonia 15	[52]	6.4	0.0	0.0	0.0	1.0	0.8	0.0	5.4	0.0	0.0	34.5	7.2	5.7
Estonia 17	[52]	15.3	0.0	0.0	0.0	1.0	1.1	0.0	5.4	0.0	0.0	33.1	6.2	1.5
Estonia 18	[52]	10.7	0.0	0.0	0.0	0.5	0.5	0.0	2.2	0.0	0.0	30.0	3.1	1.4
Estonia 19	[52]	6.9	0.0	0.0	0.0	0.8	0.8	0.0	11.7	0.0	0.0	27.6	10.5	7.4
Estonia 2	[52]	0.1	0.0	0.0	0.0	0.2	0.1	0.0	7.0	0.0	0.0	30.1	6.6	24.0
Estonia 5	[52]	0.0	0.0	0.0	0.0	0.8	0.5	0.0	3.4	0.0	0.0	32.9	4.3	25.0
Estonia 7	[52]	0.6	0.0	0.0	0.0	1.0	0.5	0.0	3.5	0.0	0.0	27.6	1.7	20.8
Estonia 10	[52]	0.0	0.0	0.0	0.0	0.9	0.3	0.0	13.3	0.0	0.0	28.4	6.8	16.4
Estonia 4	[52]	0.0	0.0	0.0	0.0	1.9	0.9	0.0	4.6	0.0	0.0	4.8	5.9	45.0
Poland 1	[53]	1.1	9.0	0.0	0.0	0.1	43.0	0.0	6.9	0.5	0.2	0.0	2.1	0.8
Poland 2	[54]	1.8	10.8	0.0	0.0	0.2	44.9	0.0	0.0	0.9	0.1	0.0	1.3	0.8
Kazakhstan 1	[46]	0.0	13.5	0.0	0.0	13.2	55.2	0.0	1.4	0.0	0.0	0.0	0.0	0.0
Kazakhstan 3	[46]	0.0	15.7	0.0	0.0	11.3	55.9	0.0	0.0	0.0	0.0	0.0	0.0	0.0
India, Kumaon	[55]	0.5	3.4	0.0	0.0	58.8	1.0	0.0	1.2	0.2	0.0	0.0	0.0	0.4
Serbia, Čačak	[56]	0.0	0.0	0.0	1.2	37.4	0.4	0.0	0.0	0.0	0.0	0.0	6.7	0.0
Kazakhstan 6	[46]	0.0	14.6	0.0	0.0	58.3	6.2	0.0	0.0	0.0	0.0	0.0	0.0	0.0
Kazakhstan 7	[46]	0.0	15.1	0.0	0.0	44.3	6.1	0.0	0.0	0.0	0.0	0.0	0.0	0.0
India	[57]	0.1	1.9	0.5	0.0	17.7	11.8	0.0	0.0	0.0	0.0	0.0	0.0	0.0
Kazakhstan 2	[46]	0.9	13.9	0.0	0.0	28.6	38.4	0.0	0.8	0.0	0.0	0.0	0.0	0.0
Kazakhstan 5	[46]	0.0	13.5	0.0	0.0	31.7	25.7	0.0	1.3	0.0	0.0	0.0	0.0	0.0
Kazakhstan 8	[46]	1.0	16.5	0.0	0.0	24.2	35.3	0.0	0.7	0.0	0.0	0.0	0.0	0.0
Estonia 1	[52]	1.4	0.0	0.0	0.0	0.0	0.1	0.0	8.6	0.0	0.0	1.7	11.2	1.6
Estonia 11	[52]	11.2	0.0	0.0	0.0	0.9	0.6	0.0	11.0	0.0	0.0	4.2	3.3	10.8
Estonia 12	[52]	0.4	0.0	0.0	0.0	2.9	3.5	0.0	13.2	0.0	0.0	2.7	12.4	17.7
Estonia 14	[52]	20.2	0.0	0.0	0.0	0.8	0.7	0.0	9.4	0.0	0.0	15.2	10.2	2.2
Estonia 16	[52]	18.6	0.0	0.0	0.0	1.3	1.3	0.0	13.0	0.0	0.0	2.2	11.4	8.2
Estonia 20	[52]	10.5	0.0	0.0	0.0	1.1	1.5	0.0	10.5	0.0	0.0	6.0	11.2	9.2
Poland 10	[47]	10.5	0.8	0.1	0.4	8.4	1.6	0.0	0.0	1.8	0.1	1.3	0.0	2.0
Poland 11	[47]	8.3	1.0	0.0	0.0	0.0	0.2	0.0	0.0	0.9	0.1	0.1	0.0	0.4
Poland 12	[47]	5.2	0.8	0.1	0.0	0.3	0.6	0.0	0.0	3.2	3.1	0.1	0.0	0.2
Poland 14	[47]	9.4	0.0	0.0	0.1	1.0	2.7	0.0	0.0	3	0.0	0.4	0.0	10.7
Poland 15	[47]	15.7	3.7	0.0	0.3	7.7	1.2	0.0	0.0	1.7	0.1	1.2	0.0	2.1
Poland 16	[47]	2.9	0.2	0.1	0.1	0.9	2.7	0.0	0.0	1.6	0.1	3.0	0.0	1.8

Ref.—reference; TS—this study; G—Gradina; B—Brzeće; bb—Belo brdo; Ž—Žitorađa.

**Table 5 molecules-25-05482-t005:** *O. vulgare* ssp. *vulgare* EO constituents from this study and others reported previously in the literature.

Locality	Ref.	Sabinene	p-Cymene	1,8-Cineole	*γ*-Terpinene	Linalool	*α*-Terpineol	*β*-Citronellol	Linalyl Acetate	Thymol	Carvacrol	*β*-Caryophyllene	Germacrene D	Caryophyllene Oxide
Serbia, Rtanj	TS	14.0	0.0	14.2	0.4	0.0	0.7	0.0	0.0	0.0	0.0	13.4	21.5	3.9
Montenegro, Radovče	[62]	3.6	1.2	3.2	0.6	4.9	5.4	0.0	0.6	1.2	0.2	14.4	27.9	1.8
Montenegro, Lipovo	[62]	5.9	4.6	2.9	1.4	6.6	4.8	0.0	0.0	0.2	1.0	12.7	15.4	2.2
Montenegro, Grahovo	[62]	4.6	3.7	3.3	2.8	3.1	5.7	0.0	0.5	0.0	0.0	14.6	16.8	2.7
Moldova	[63]	9.8	0.7	0.3	2.1	0.8	0.6	0.0	0.0	0.3	11.7	13.1	17.0	1.6
Lithuania	[64]	7.4	0.2	8.5	0.1	11.3	2.7	0.0	0.0	0.2	0.1	10.0	13.2	3.4
Albania, n3	[64]	7.3	0.6	3.4	1.8	0.3	2.3	0.0	0.0	2.6	0.2	11.1	18.6	1.2
Lithuania, Silenai	[65]	8.7	0.2	6.1	1.9	1.4	1.5	0.0	0.0	0.0	0.0	12.7	14.1	2.1
Lithuania, Rastinenai	[65]	10.1	0.8	6.4	1.6	1.8	1.5	0.0	0.0	0.0	0.0	12.6	14.2	4.4
China, Kunlun Mt	[66]	0.0	0.0	0.0	0.0	0.4	0.0	85.3	0.0	0.0	0.0	0.4	0.0	0.0
Pakistan	[66]	0.0	0.3	0.0	0.0	0.4	0.0	72.7	0.0	7.2	0.0	1.0	0.0	0.4
China, Hetian	[66]	0.0	0.0	0.0	0.0	0.5	0.0	75.0	0.0	0.0	0.0	0.5	0.2	0.5
China, Shangqiu	[66]	0.0	7.4	0.0	1.9	0.4	0.0	0.0	0.0	42.9	0.0	7.8	0.3	2.2
Iran, Noshahr	[67]	0.8	3.6	3.8	9.7	0.3	1.2	0.0	0.0	37.1	9.6	1.9	1.1	0.7
China, Anhui	[66]	0.5	0.9	20.8	0.4	5.5	0.0	0.0	0.0	1.5	0.0	10.2	0.5	2.5
Czech	[64]	6.3	1.1	17.4	0.0	2.0	1.7	0.0	0.3	0.0	0.0	6.0	6.1	13.5
China, Yili	[66]	0.0	0.0	0.0	0.0	3.2	3.9	0.0	0.0	0.0	0.0	17.7	9.8	32.9
Iran, MeshkinSahahr	[68]	2.2	0.0	1.4	12.3	0.0	0.0	0.0	0.0	0.0	8.0	1.2	0.0	21.0
Finland	[64]	2.3	7.2	3.3	3.7	0.0	0.3	0.0	0.0	0.0	4.6	7.9	5.5	11.4
Montenegro, Boljevići	[62]	0.9	2.0	0.8	3.1	8.8	17.8	0.0	9.7	8.3	1.1	7.7	16.0	0.5
Spain	[64]	0.4	0.0	0.0	0.0	3.2	52.8	0.0	0.0	0.0	0.0	6.4	11.3	1.3
Montenegro, Kameno	[62]	0.0	7.8	0.0	6.5	0.0	0.2	0.0	0.0	0.0	74.3	1.3	0.2	0.0
Iran, Kaleybar	[68]	0.9	0.0	0.0	17.5	0.0	0.0	0.0	0.0	0.0	21.3	11.3	1.5	2.7
Croatia	[64]	0.0	2.6	0.0	1.6	0.0	0.1	0.0	0.0	0.3	86.0	0.4	0.0	0.2
Portugal, n7	[64]	0.4	5.1	0.0	9.8	0.6	0.1	0.0	0.4	0.3	26.5	9.3	4.6	2.9
Iran, Gardrahmat	[69]	0.0	0.0	0.0	20.5	0.0	0.0	0.0	0.0	15.4	23.5	5.1	0.0	2.1
Iran, Khezr Abad	[69]	0.0	0.0	0.0	18.4	0.0	0.0	0.0	0.0	0.0	59.4	0.0	0.0	0.0
Iran, Perdanan	[69]	0.0	0.0	0.0	9.6	0.0	0.0	0.0	0.0	0.0	58.5	3.7	0.0	0.3
Iran, Garderan	[69]	0.0	0.0	0.0	7.7	0.0	0.0	0.0	0.0	0.0	67.1	0.0	0.0	0.4
Iran, Sabalan	[68]	20.8	0.0	0.0	7.1	0.0	0.0	0.0	0.0	0.0	17.8	6.1	0.0	3.1
Poland, Drohiczyn 1	[70]	25.4	1.7	0.9	3.3	2.7	0.8	0.0	0.0	2.1	0.5	11.5	14.2	3.0
Poland, Radecznica	[70]	25.9	2.1	2.2	2.5	3.4	1.7	0.0	0.0	1.1	1.8	14.8	11.2	2.4
Poland, Lupkow	[70]	14.5	7.4	1.5	6.1	3.3	1.3	0.0	0.0	1.7	1.7	10.8	11.8	1.3
Poland, Javornik	[70]	19.9	2.3	11.9	3.7	6.0	3.5	0.0	0.0	1.1	1.9	9.7	8.4	1.0
Poland, Zwierzyniec	[70]	17.1	4.4	10.0	5.6	4.0	1.7	0.0	0.0	0.5	0.1	9.8	2.9	10.5
Poland, Babice	[70]	15.4	5.1	8.3	1.5	14.1	3.2	0.0	0.0	2.0	2.2	10.1	0.6	3.4
Poland, Czarnystok	[70]	21.6	11.0	4.5	6.5	4.9	0.7	0.0	0.0	0.6	0.6	8.3	7.2	3.7
Poland, Monasterzec	[70]	15.0	2.9	2.3	8.3	10.4	1.7	0.0	0.0	2.0	1.3	11.8	5.1	2.7
Iran, Chalus	[68]	3.2	0.0	2.8	16.5	0.0	0.0	0.0	0.0	0.0	6.4	1.4	0.0	0.0
Poland, Lukowe	[70]	12.1	1.5	14.7	1.8	1.1	4.6	0.0	0.0	0.9	2.0	18.2	9.6	2.1
Poland, Drohiczyn 2	[70]	6.2	12.7	1.6	1.6	4.1	1.9	0.0	0.0	1.0	1.9	21.3	1.9	9.0
Latvia	[64]	0.0	0.0	0.0	0.0	0.0	0.0	0.0	0.0	0.0	0.0	25.1	19.4	13.1
Turkey	[71]	1.6	0.1	2.1	0.6	1.9	2.2	0.0	1.9	0.0	2.3	20.9	17.8	0.0
Poland, Lipowiec	[70]	6.5	8.6	8.9	0.2	32.1	3.3	0.0	0.0	1.4	0.7	3.5	0.2	10.0
Poland, Karlikow	[70]	8.1	8.5	7.9	1.4	15.0	3.8	0.0	0.0	0.8	1.5	17.8	1.5	3.9
Poland, Bukowsko	[70]	1.9	5.9	0.2	5.7	24.2	1.1	0.0	0.0	1.2	1.6	19.6	5.9	2.4
Poland, Gruszka	[70]	7.4	4.5	0.4	6.2	16.0	1.2	0.0	0.0	1.5	0.8	10.1	4.7	2.5
Portugal, n8	[64]	0.1	0.2	0.0	0.6	84.7	0.3	0.0	0.0	0.0	4.7	1.5	2.4	0.1
Albania, n7	[64]	2.3	0.4	1.5	1.0	1.3	0.8	0.0	33.0	2.8	0.2	9.0	20.6	0.3
Iran, Ilam	[72]	1.5	0.0	5.4	0.9	1.4	0.0	0.0	0.0	25.3	12.3	0.0	0.1	0.0
India, Srinagar	[73]	15.3	23.9	0.0	38.4	0.0	0.0	0.0	0.0	0.0	0.9	1.2	1.2	0.0
India, Pulwama	[73]	18.1	21.1	0.0	25.7	0.0	0.0	0.0	0.0	0.1	14.8	3.0	1.7	0.0
India, Tangmarg	[73]	6.5	33.3	0.0	8.4	0.0	0.0	0.0	0.0	0.8	27.2	0.9	0.6	0.0

Ref.—reference; TS—this study.

**Table 6 molecules-25-05482-t006:** *A. millefolium* EO constituents from this study and others reported previously in the literature.

Locality	Ref.	Sabinene	*β*-Pinene	1,8-Cineole	Camphor	Borneol	Terpinene-4-ol	Bornyl Acette	chrysanthenyl Acetate	Lavandulyl Acetate	*β*-Caryophyllene	δ-Cadinene	Elemol	chamazulene
India, Himachal Pradesh	[83]	17.6	6.3	13.0	0.0	12.4	6.2	8.0		0.0	2.3		0.0	5.3
Serbia, Busilovac	[84]	16.9	5.1	10.1	7.1	1.6	12.9	0.0	3.0	3.4	3.3	0.2	0.0	0.4
Serbia, Sokobanja	[84]	1.1	21.3	7.3	3.7	19.5	1.8	0.1	0.0	0.0	2.8	0.7	0.6	0.0
Serbia, Sokobanja2	[84]	0.9	13.2	11.1	0.5	6.7	1.1	0.0	0.0	0.2	2.5	0.7	9.9	0.0
Serbia, Ražanj	[84]	2.0	16.3	7.8	1.8	0.6	0.5	0.0	0.0	0.3	3.4	1.6	14.7	0.0
Serbia, Ravna Gora	[84]	4.6	23.6	16.9	0.4	0.0	0.6	0.0	0.0	0.0	8.3	0.2	0.0	10.5
Serbia, Topli do	[84]	4.5	26.7	17.1	1.0	2.8	0.5	0.0	0.0	0.0	9.4	0.3	0.0	7.4
Serbia, Dobro polje	[84]	3.4	36.3	18.4	1.1	0.0	0.9	0.0	0.0	0.0	5.9	0.2	0.0	1.7
Serbia, Rajac1	[84]	5.7	18.7	10.8	1.3	0.0	0.8	0.0	0.0	0.0	16.0	0.0	0.0	1.3
Serbia, Medvednik	[84]	4.9	24.8	10.5	3.9	1.7	0.6	0.0	0.0	0.0	5.7	0.5	1.1	10.1
Serbia, Rajac2	[84]	3.0	28.6	11.7	11.3	5.1	1.2	0.0	0.0	0.0	3.1	0.1	1.1	2.1
Serbia, Ovča	[84]	2.0	28.2	11.7	1.8	2.2	0.9	0.0	0.0	0.5	7.9	0.4	0.1	1.7
Lithuania1	[85]	1.7	14.2	8.0	4.9	3.8	3.9	1.6	0.0	0.0	7.1	1.8	0.0	4.1
Lithuania2	[85]	2.2	14.0	7.8	3.8	4.9	1.8	1.2	0.0	0.0	5.3	0.8	0.0	2.8
Lithuania3	[85]	0.9	15.5	10.1	2.8	2.3	3.6	2.0	0.0	0.0	5.7	1.0	0.0	3.4
Lithuania4	[85]	2.1	15.2	11.8	2.8	3.9	1.7	2.1	0.0	0.0	5.6	0.9	0.0	2.7
Lithuania5	[85]	7.0	10.2	7.7	0.6	0.4	0.9	1.2	0.0	0.0	7.5	2.8	0.0	4.8
Lithuania6	[85]	7.2	17.2	9.3	4.7	2.5	5.3	0.6	0.0	0.0	3.7	0.7	0.0	0.0
Lithuania7	[85]	6.5	15.6	6.7	6.6	4.3	7.6	0.9	0.0	0.0	3.0	1.0	0.0	0.0
Lithuania8	[85]	13.0	13.6	9.4	3.6	1.5	5.9	1.2	0.0	0.0	3.7	1.1	0.0	0.0
Lithuania16	[85]	1.9	12.3	6.4	3.5	7.6	2.2	3.3	0.0	0.0	5.8	1.4	0.0	0.0
Lithuania17	[85]	1.5	12.1	3.1	3.6	8.0	2.0	3.6	0.0	0.0	1.9	0.8	0.0	0.0
Lithuania18	[85]	7.1	9.1	6.4	1.8	2.7	2.0	2.2	0.0	0.0	3.2	1.5	0.0	0.9
Lithuania19	[85]	5.4	7.0	4.5	0.9	3.7	1.6	1.6	0.0	0.0	5.5	1.9	0.0	0.0
Serbia, Sokobanja4	[84]	6.1	8.1	12.9	3.0	6.6	3.5	0.0	0.1	0.1	1.9	0.5	3.9	0.0
Serbia, Sokobanja 5	[84]	2.6	12.6	14.5	3.8	0.9	0.5	0.0	0.0	0.0	8.3	0.3	0.0	2.2
Serbia, Aleksinac1	[84]	8.0	3.6	10.4	1.1	0.0	3.0	0.0	0.0	0.3	4.0	0.4	7.6	0.0
Serbia, Aleksinac2	[84]	12.2	2.9	17.3	1.1	1.3	2.6	0.0	0.0	0.0	2.5	0.3	15.5	0.0
Serbia, Vojska	[84]	10.2	0.8	14.3	1.1	1.7	6.5	0.0	0.0	0.3	0.8	0.2	12.5	0.0
Serbia, Velika Plana	[84]	6.1	3.9	13.6	1.4	13.5	1.9	0.0	0.0	0.8	3.3	0.4	4.8	0.9
Serbia, Begaljica	[84]	6.9	19.6	26.6	2.6	9.1	1.2	0.0	0.0	0.0	7.4	0.2	0.1	0.0
Italy, Cagliary	[86]	14.6	1.1	17.2	1.6	1.6	5.7	0.0	0.0	0.0	2.3	0.0	0.0	0.1
Iran, Chaharbagh	[87]	3.2	2.1	18.6	13.9	9.4	1.2	1.3	0.0	0.0	2.1	0.0	0.0	2.3
Turkey, Sivaz	[88]	2.8	4.2	24.6	16.7	4.0	2.8	0.1	0.0	0.0	0.4	0.0	0.0	0.0
Lithuania9	[85]	4.5	7.3	9.6	5.9	2.4	1.7	1.6	0.0	0.0	2.6	1.3	0.0	1.2
Lithuania10	[85]	3.7	9.5	9.9	2.0	2.0	1.9	4.7	0.0	0.0	2.6	0.6	0.0	0.0
Lithuania11	[85]	2.5	6.6	8.8	2.4	2.5	1.3	3.7	0.0	0.0	6.5	1.0	0.0	0.8
Serbia, Rtanj	TS	2.8	1.5	5.6	9.8	1.6	6.3	0.4	0.4	0.0	4.7	0.0	0.7	0.1
France, Toulouse	[89]	6.7	3.4	4.0	12.8	1.8	4.7	1.2	0.0	0.0	1.7	1.2	1.6	0.0
Lithuania15	[85]	3.6	4.5	8.8	13.1	12.8	2.2	0.7	0.0	0.0	0.7	0.3	0.0	0.0
Serbia, Sokobanja3	[84]	1.9	15.4	14.3	0.9	20.2	1.2	0.0	0.0	0.1	2.5	0.2	8.4	0.0
Lithuania12	[85]	4.3	6.5	9.5	4.1	11.5	2.6	1.1	0.0	0.0	3.5	0.6	0.0	0.7
Lithuania13	[85]	1.5	4.0	5.3	7.2	13.1	1.9	1.9	0.0	0.0	1.9	0.6	0.0	1.3
Lithuania14	[85]	3.1	12.6	12.5	7.2	13.2	4.5	0.5	0.0	0.0	2.5	0.5	0.0	0.0
Serbia, Secanj	[90]	2.8	3.2	0.1	2.0	1.4	0.0	0.0	21.3	0.9	9.5	0.0	0.0	1.5
Serbia, Ćuprija	[84]	5.6	5.6	2.1	9.4	3.1	1.1	0.0	0.0	11.0	5.3	0.4	1.1	0.2
Serbia, Aradac	[90]	3.6	4.1	3.5	1.0	5.3	0.0	0.0	6.8	14.9	7.6	0.5	0.0	13.9
Serbia, Suvobor	[84]	1.2	10.1	2.0	0.2	0.0	0.8	0.0	0.0	0.0	12.1	0.8	0.0	3.4
Turkey, Elazig	[91]	2.7	0.3	2.3	0.0	0.0	0.0	0.0	0.0	2.3	4.9	19.0	0.6	0.0
Serbia, Maljen	[84]	8.9	22.2	1.9	0.1	0.1	1.3	0.0	0.0	1.6	13.9	0.6	0.0	29.1
Estonia	[84]	6.6	17.6	7.6	0.3	0.4	0.2	0.4	0.0	0.0	9.8	1.2	0.0	30.7
UK	[92]	4.5	11.5	4.9	4.3	1.8	0.4	0.7	0.2	0.0	6.6	1.1	0.0	25.8
Latvia	[92]	4.0	12.0	4.7	1.7	1.5	0.2	1.5	0.1	0.0	6.7	1.3	0.0	26.8
Norway	[92]	2.7	9.3	2.9	1.1	0.9	0.3	1.1	0.0	0.0	6.3	1.3	0.0	40.2
Lithuania20	[85]	3.2	15.1	6.4	0.5	1.2	2.3	0.0	0.0	0.0	8.0	2.1	0.0	20.1
Serbia, Divčibare2	[84]	3.1	10.2	18.1	0.9	0.8	0.8	0.0	0.0	0.0	5.1	0.3	0.0	22.2
Ukraine	[92]	2.0	5.9	6.0	1.2	1.2	1.7	0.7	0.1	0.0	4.3	1.7	0.0	23.7
Germany	[92]	0.0	0.1	0.0	0.1	1.4	1.2	7.3	0.0	0.0	2.5	1.2	0.0	44.3
Austria	[92]	4.9	7.7	2.5	0.9	1.7	3.1	15.8	0.0	0.0	3.6	0.5	0.0	15.7
Serbia, Padej	[84]	6.3	0.9	1.7	0.3	0.0	0.6	0.0	0.8	18.1	3.2	0.3	0.0	23.1
Serbia, Divčibare1	[84]	0.6	11.0	4.5	0.3	0.0	0.4	0.0	0.0	8.2	15.8	0.2	0.1	18.3
Serbia, Slano Kopovo	[90]	5.3	8.9	5.3	1.5	1.0	0.0	0.0	5.8	0.9	9.0	0.0	0.0	15.8
Czech	[92]	0.0	0.0	0.8	0.7	0.6	0.5	1.2	0.0	0.0	9.7	1.9	0.0	25.8

Ref.—reference; TS—this study.

**Table 7 molecules-25-05482-t007:** *A. clypeolata* EO constituents from this study and others reported previously in the literature.

Locality	Ref.	Camphene	Sabinene	*β*-Pinene	p-Cymene	1,8-Cineole	Camphor	Borneol	Terpinen-4-ol	*α*-Terpineol	allo-Aromadendrene	Caryophyllene Oxide	epi-*α*-Cadinol	*α*-Cadinol
Serbia, Rtanj	TS	2.3	0.3	1.4	3.1	45.1	18.2	2.7	2.8	2.4	1.3	3.2	2.4	1.5
Serbia, Rtanj	[95]	1.9	0.0	0.8	3.6	38.6	19.9	3.6	6.5	5.3	0.4	1.7	1.9	1.1
Serbia, Rudina	[96]	0.0	0.0	0.0	0.0	16.0	9.2	11.9	8.8	2.4	0.0	11.5	0.0	0.0
Italy, Turin	[97]	1.0	9.2	23.7	1.8	10.1	2.2	1.2	2.6	2.6	0.0	0.8	0.0	0.0

Ref.—Reference; TS—This study.

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
