# Peer review of "Chemical Characterization and Antibacterial Activity of Essential Oil of Medicinal Plants from Eastern Serbia"

_molecules, 2020, doi:10.3390/molecules25225482_

Round 1
Reviewer 1 Report
Aćimović et al., in the manuscript entitled “Chemical Characterization of Essential Oil and Antibacterial activity of Five Medicinal Plants from Eastern Serbia”, aimed at characterizing the essential oil of five medicinal plants as well as at analyzing the antibacterial activity.
In my opinion, while the chemical characterization of the essential oil, as well as the comparison with the literature data, are accurate, the tests carried out to analyze the antibacterial activity should be implemented.
This paper has the potential to be accepted, but several points below listed have to be clarified or fixed before positive action can be taken.
- The title is confusing, I suggest to change it as follow: “Chemical Characterization and Antibacterial activity of Essential Oil of Five Medicinal Plants from Eastern Serbia”
- Introduction section lines 35-37: authors should include different references for the mentioned topics (flavoring, food preservation, medicinal purposes) and also specify what do they mean by “bioactive properties”.
- Introduction section lines 50-52: authors should include different references since they mention “a number of reports” and expand this paragraph including studies on antimicrobial activity of Essential Oils (EOs).
- Methods section: paragraph 2.2 - EO extraction, should be revised including more accurate information on the isolation process.
- Information on the statistical analysis carried out for the antibacterial activity data is missing and has to be included. The criteria of this analyses have to be specifically reported in Methods section, paragraph 2.6 - Statistical analysis.
- In Table 1, the authors should separate volatile and non-volatile compounds.
- In Table 2, MIC and MBC abbreviations should be explained in a legend. Also, authors should indicate how many times the experiment was carried out, including in the table the mean with the standard deviation as well as the statistical significance.
- The experiments carried out to test the antibacterial analysis should be implemented with other tests.
- Information on the EO concentration used in the antibacterial activity experiments is missing; also, the results from more than one concentration should be included. It could be useful to analyze if higher concentrations lead to faster bacterial death, revealing a dose-dependent effect.
- Scanning Electron microscopy (SEM) could be useful to observe the morphological changes resulting from treatment with Essential Oil of Five Medicinal Plants from Eastern Serbia. Aline Cristina Guimarães et al., in the paper “Antibacterial Activity of Terpenes and Terpenoids Present in Essential Oil” observed that the untreated E. coli had bacillary form and smooth surfaces, whereas the treated cells were irregularly sized with the presence of debris, possibly by disrupting cell division or dysfunction of the cellular membrane (https://www.ncbi.nlm.nih.gov/pmc/articles/PMC6651100/#!po=22.000, Antibacterial activity and mechanism of cinnamon essential oil against Escherichia coli and Staphylococcus aureus, Yunbin Zhang et al., 2015).
- The authors reported that aureus was more sensitive to the tested EOs than E. coli. However, S. Kitaibeli expressed the most potent activity against both G-positive and G-negative bacteria, followed by T-serpyllum and A. clypeolan, while O. vulgare ssp. Vulgare showed the weakest antibacterial activity. They in paragraph 3.1 reported the chemical composition of EOs studied, showing in particular, the main EO constituents, probably responsible for the biological effect observed. It could be useful to study the antibacterial activity of EOs focusing on their major constituents to study and in the future concentrate if necessary one of the components that could be responsible for the observed effect. The time-Kill curve analysis was used to characterize the bactericidal action, in this way the authors could evaluate the kinetics of the compounds killing the two bacterial strains and if it were concentration-dependent (for example EO from O. Vulgare ssp. Vulgare, A. clypeolata, A. millefolium contain different amount of 1,8-cineole; EO from A. millefolium and A. clypeolata contain different amount of camphor). Moreover, this analysis could be useful to understand the particular activity of A. millefolium that was very potent against S. aureus.
- Moreover, the introduction of other G-positive and G-negative bacteria could be useful to confirm the obtained data and the more specific effect of A. millefolium.
- In literature several data showing the antibacterial activity of EOs extracted from the same five Medicinal Plant analyzed in this study are reported. The specificity of the results reported in this study concerns the geographic origin of the plants. Thus, it could be interesting to compare the effect of the EOs of the five Medicinal Plants from Eastern Serbia with those derived from the same plants from other regions to highlight differences in their activity. Moreover, to add value to the presented data, beside to evaluate the effects of EOs individually, I suggest to assy them in combination with standard antimicrobials to highlight eventual synergistic interactions.
- Since the authors conclude that “The results can be utilized by the perfumery and cosmetics, food, and pharmaceutical industries”, suggesting the use of EO for human purposes, data on the concentration of EO to be used on human cell lines should be included in the manuscript. For example, authors could perform cell viability and/or cell toxicity assays of EO on in vitro human cell lines to determine a non-toxic concentration range for each EO.
Author Response
The authors would like to thank the Editor and Reviewers for a quick and professional review. It is obvious that the Reviewers are experts in this field. All the Reviewers' remarks are accepted and paper is changed according to their comments. The authors believe that the changed paper would satisfy the Reviewers' criteria and that it is going to be interesting enough for publishing in the Molecules journal.

Reviewer 2 Report
The manuscript deals with the chemical characterization of essential oil and antibacterial activity of five medicinal plants from eastern Serbia.
Title- Please remove “five”.
The English language must be revised.
Please format the scientific names in italic.
Please format units in accordance, “37 ºC” not “37ºC”.
Introduction
This section is too short and must be improved.
Materials and methods
Line 71- “Clevenger apparatus was used to extract the essential oil from air-dried aboveground parts of each sample for 3 h. This process was performed in three replications, and the obtained EOs were dried over anhydrous NaSO4 and stored in dark at 4 °C for further analysis.”??This method should be presented in more detail.
Results and discussion
Line 123- “According to the assay, S. aureus was more sensitive to the tested EOs than E. coli. However, S. kitaibelii expressed the most potent activity against both G-positive and G-negative bacteria, followed by T. serpyllum and A. clypeolata, while O. vulgare ssp. vulgare showed the weakest antibacterial activity. A. millefolium showed weak activity against E. coli, but was very potent againstS. aureus (Table 2).”??Table 2 shows MBC values >20 µL/mL for E. coli using O. vulgare ssp. vulgare and A. millefolium. Which concentration is higher than 20 µL/mL that is the lowest concentration required to kill the bacteria??since in the other experiments 20 µL/mL was enough.
Conclusion
This section must be improved.
References
Around 48 references have more than 5 years. Please update your list of references.
Please format each reference according to the guide for authors.
Author Response

(The authors gave the same response as above.)

Round 2
Reviewer 1 Report
Even if the authors responded properly to the majority of the comments, some points remain not resolved since the authors have planned future investigations for studying the suggested aspects. Thus, I think it would be opportune to add sentences about this intention within the text. This refers to points 8, 10-14.
Author Response
Dear reviewer,
we thank you for highly constructive suggestions which have significantly contributed to quality of this paper.
Sincerely,
Authors
Reviewer 2 Report
The manuscript was improved.
Author Response

(The authors gave the same response as above.)
